# Evaluation of Materials and Structures with a Multistatic Ultra-Wideband Impulse Radar: A Concept Validation

Gatis Gaigals [1,*], Romans Maliks [1], Vladimir Aristov [1], Rolands Savelis [1], Janis Simanovics [1], Eduards Lobanovs [1], Haralds Egliens [1], Dans Laksis [1], Kristaps Maris Greitans [1] and Modris Greitans [2]

1 Signal Processing Laboratory, Institute of Electronics and Computer Science, LV-1006 Riga, Latvia
2 Institute of Electronics and Computer Science, LV-1006 Riga, Latvia
* Correspondence: gatis.gaigals@edi.lv

**Abstract:** This document describes the results of the study contributing to the methods and tools applicable in plastic waste sorting systems that exploit the multistatic ultra-wideband impulse radar enforced with a deep learning signal processing back-end. The novelty of the research is the use of synthetic data for the development of a trained neural network before real data are available, and the use of a multistatic radar for the improvement of the training data set. The study results are described in multiple publications; the current paper shows the applicability of the described approach. The main results are as follows: a monostatic impulse radar can be used for the determination of material properties, such as thickness, dielectric permittivity, and losses, with limited accuracy; multistatic radar configuration increases the accuracy of the material property estimation; an open source finite difference time domain simulator can be used to simulate electromagnetic wave propagation in dielectric structures in order to generate synthetic data for development of optimized artificial neuron network structures used for the estimation of dielectric material properties, and the developed network can successfully be used for multistatic radar data processing.

**Keywords:** multistatic radar; ultra-wideband radar; radar measurements; radar signal processing; multidimensional signal processing; data preprocessing; transforms; artificial neural networks





## 1. Introduction

The world sinks in waste—the ability of humankind to reduce the pollution of the world with industrial exhaust gases and household waste can determine the ability of humans to end the Anthropocene, i.e., the ongoing extinction initiated by humans [1]. One of the least visible and therefore one of the most dangerous pollutants is plastic. Plastics can be found on the ocean floor [2] and in marine animals [3]; microplastics pollute drinking water supplies [4] and even unborn babies [5,6].

In order to reduce plastic pollution is the recycling of plastic waste. In the process of recycling, the most important step is the sorting of waste, since there is a lot of different types of plastic that have to be sent to different recycling companies. One way to sort plastic and waste in general is the application of machine vision [7,8]. Such systems are rather expensive, since the hardware used for the processing of visual information is intensely computing oriented. A result of such computing is the determination of markers that are compared to the markers of known products (not materials, not types of plastic) that contain a certain type of plastic. Systems based on machine vision have two major disadvantages: (1) they can only sort known products, if there is no information in the marker database on a new type of waste, it can not be classified; (2) if a known product is packed in a visually similar but different type of plastic packaging it can be classified incorrectly if the machine vision does not determine the type of plastic from the triangle marker on the package; this is a serious problem for pressed waste because the marker can be damaged on the way to the sorting site.

The study described in this article contributes to the methods applicable in waste sorting systems that exploit the multistatic ultra-wideband (UWB) impulse radar (IR) enforced with a deep learning approach in combination with conventional signal pre/postprocessing methods. The project was based on two facts: (1) the propagation of electromagnetic (EM) waves does not depend on the color of the waste, but on the form and the material only; (2) the reflected or propagated-through-material EM wave contains information about properties of the material that in general is very hard to decode (deconvolute using multiple kernels) using exact signal processing methods; instead, artificial neuron networks (NNs) can be trained to recognize the material under analysis, since NNs have proven their value in different classification tasks [9–11], and the theory is well developed [12–14].

The idea to combine radar and NN is not new, and it has been previously conducted for different needs: human activity recognition [15], gesture recognition [16], healthcare [17,18], object classification [19], and localization [20]. Nevertheless, this study is unique since, at the time of writing this article, only a few publications exist that are devoted to the determination of material properties by means of UWB radar and NNs [21–24]. The drawbacks of the referenced system are that objects under analysis have to be in contact with the radar [22]; it has only been validated using simulated data [23,24]; and it can only determine irregularities in the radiated objects [21], and not exact material properties.

This study investigated the hypothesis that multistatic radar architecture in combination with the use of NN-based signal processing can provide an opportunity to determine different properties of the radiated object: relative dielectric permittivity, loss tangent, material thickness, and object type (glass bottle, plastic bottle, metallic can) and shape (whole or smashed object).

## 2. Materials and Methods

In order to complete the study, it was necessary to complete the following goals:

- Upgrade the existing EDI pseudo-monostatic UWB IR radar [25] to a multistatic system in order to validate theory with results of experiments;
- Develop a model that describes the passing of the multistatic UWB IR signals through a layered medium of various configurations, and that could be used to generate correct synthetic NN training data sets;
- Develop signal preprocessing methods based on multidimensional representations (time frequency, time scale, etc.) and signal decomposition, combining signals reflected from and propagated through a medium in order to ensure improved accuracy and/or reduced complexity of the NN-based analysis;
- Research favorable NN architecture (recurrent NN, long short-term memory NN, pulse-coupled NN, transformers, convolution NN (CNN), etc.) and its enhancement for optimal (precision, complexity, etc.) estimation of object material and its structure.

### 2.1. Ultra-Wideband Impulse Radar

EDI has a self-designed UBM IR in pseudo-monostatic configuration with a main power of impulse in the frequency region from 1 to 3 GHz and a peak in the power spectrum at 1.5 GHz [25]. For our study needs, this had to be upgraded to multistatic configuration, extending the frequency range as wide as possible in order to capture imprints of radiated object properties in received signals.

### 2.2. Synthetic Data Generation

The main aspect that determined the use of synthetic data for our study of NN capabilities was the lack of objects available for measurement: it is unrealistic to gather many objects of different thickness (varying from 200 to 500 mm), different relative dielectric permittivity $\epsilon_r$ (varying from 1 to 8), and different absorption $\tan\delta$ (varying from 0 to 0.1) in order to evaluate NN performance using real measurement signals.

At the start of the study, the team did not have any reliable tool or method for EM wave propagation synthetic data generation; thus, at the start, the UWB IR EM wave propagation model had to be developed and tested.

Meanwhile, the EM wave propagation simulationpars software had to be tested in order to determine whether its simulated signals of impulse radar scenes corresponded to the signals of the real impulse radar. For this purpose, the open source EM field solver OpenEMS was chosen [26]. For the simulation of EM wave propagation, it uses the finite difference time domain (FDTD) method, which is mandatory in the case of impulse radars. For the needs of synthetic data generation, the simulation models of the EDI radar signal and antennas had to be developed.

To estimate dielectric material properties with the addition of AI, the regression problem must be solved. In machine learning, the regression problem is characterized as prediction calculus. Unlike the classification problem, the regression problem occurs when numerical value prediction of a parameter is required. In our case, if we take raw data, the synthesized UWB signals with an input length of 1024 data points from a single channel are determined as input features of the NN model. Then, the linear regression implements these data as inputs for the NN, and therefore, is defined in terms of a linear function [27].

In our research, multiple inputs (which feature further in the text) are simply UWB signal data (echoed signal data points gathered on the receiving antenna/s in the simulation) of raw or any other representative form. Single output refers to the single output layer, excluding any activation, and with the numerical value as a prediction of the parameter of $\epsilon_r$ or $tan\delta$.

By gathering all features, for example, into a vector $\mathbf{x} \in R^k$, and all weights into a vector $\mathbf{w} \in R^k$, assumptions can be expressed as

$$\widetilde{y} = \mathbf{w}^\top \mathbf{x} + b \tag{1}$$

In Expression (1), the $\mathbf{x}$ corresponds to the features of a single UWB signal data point. To refer to the features of the entire data set of $n$ examples, the design matrix is used as $\mathbf{X} \in R^{n \times k}$. The $\mathbf{X}$ contains one row for every signal and one column for every feature (data point) of the signal. For the collection of features $\mathbf{X}$, the predictions $\widehat{\mathbf{y}} \in R^n$ are expressed via the matrix–vector product:

$$\widehat{\mathbf{y}} = \mathbf{X}\mathbf{w} + b, \tag{2}$$

where broadcasting is applied during the summation [28]. To sum up, the goal of the linear regression problem is to find the weight vector $\mathbf{w}$ and bias $b$ so that the features of the new and unseen UWB signal data will be predicted with the lowest error possible. The weights in the presented linear regression model are the coefficients of the independent variables that determine the slope of the line for each feature. These coefficients learn, during the training process, to make predictions and minimize the difference between the predicted and the true output.

*Convolution Neural Network Model*

During the research, several problems were defined for successful prediction of material dielectric properties with the help of the NN:

A    Choice of the NN architecture;
B    Selection of raw signal preprocessing methods;
C    Methodology of the model development and its hyperparameter tuning;
D    Selection of loss function;
E    Impact of noise.

The convolution neural network (CNN) was chosen to solve the regression problem (A) as it is used in common and powerful architectures such as LeNet5, AlexNet, and ResNet: this decision was based on results derived when different NN architectures were studied within the same research project [29,30]. Several hidden/deep non-linear layers, as well as a convolution layer, were added to improve performance of the model [31,32]. With the

results observed, the CNN showed a preferable learning rate, with only insignificant loss in accuracy performance. Other architectures showed either worse performance or drastically increased training time [29].

The impact of data preprocessing (problem (B)) is crucial to reduce the number of trainable weights and training time, and to improve the prediction accuracy on the evaluation data [33]. In this study, the following data preprocessing methods were tested:

- Cleaned time domain data;
- Principal component analysis (PCA);
- Amplitude spectrum;
- Phase spectrum;
- Short-time Fourier transform (STFT) [34].

The created data sets with different representations were used to train the proposed CNN architectural model to recognize the dielectric properties of the material, the thickness, as well as the type and structure of the objects in [29,30]. For better comparison, the evaluation/test data sets with identical data signals during the training process on different data representations were chosen. The rest of the data were divided into a 85%/15% ratio, with 85% designated to the training data set, while 15% of the data designated to validation (chosen randomly for each run). In total, 15 runs are performed for every data representation to determine the average performance for lower parameter prediction error.

During synthesized data preprocessing, input data dimensions vary, which increases the complexity of the comparison between different NN models. The input data structure varies, and based on the raw input data from a single receiver is the UWB radar signal data, with a length of 500 ($\times$2 the length of the data for bistatic and $\times$3 for multistatic setup) most significant readings. The data from a single receiver in spectrum representations consist of 200 features, whereas PCA has only 20. All of the data are one-dimensional, while the STFT data are two-dimensional [34]. The output of the neural network is a single value that corresponds to the predicted output of the material parameter (dielectric permittivity or loss tangent). In many cases, the NN model layers and hyperparameters are chosen by the trial-and-error method, which is often time-consuming and may lead to false assumptions for the best performance. To solve problem (C), the *"Grid search"* method was chosen to increase the performance of the NN and to propose the tuning algorithm for the model and hyperparameters [35–37]. The tool built in the *"Tensorflow keras"* module was used to determine parameters. With the learning rate and batch size determined, the loss functions were studied to propose the favorable function for further model training.

The loss function is one of the crucial parameters of the neural network model; it quantifies the distance between the actual and predicted value of the target data. Mean absolute error, mean squared error (MSE), and custom loss function in the form of root mean squared error were evaluated, as they are commonly used to examine regression problems [38]. Therefore, the transmitted synthetic UWB signal data were used to form data sets of different representations with a single-parameter-varying to solve problem (D) (single-parameter/two-parameter-varying data sets with only single/two parameters being non-constant for every ultra-wideband signal or its representation data were used).

In addition, the tuner tool was used to implement additional layers into the model to increase performance by tuning the hyperparameters for each individual layer. Furthermore, the tuner tool was used to search through the layer hyperparameters to reduce the validation loss value to more favorable figure of merit. At the end, an identical structure was used for training the model, but with different types of preprocessed data and input shapes (see Figure 1).

Finally, with the proposed NN model's architecture methodology, chosen preprocessing methods to study, and the prepared data sets, the following steps were carried out to achieve results on parameter evaluation:

- Introduction of single-parameter-varying ($\epsilon_r$) data sets to the CNN to define favorable loss function.

- Introduction of two-parameter-varying data sets to evaluate impact on the parameter prediction error with different antenna setups and data representations, and to determine favorable antenna setup signal and data representation.
- Addition of the Gaussian noise of different SNRs to the clearly monostatic case to test the NN model and study the impact of noise on prediction accuracy, and to determine the threshold of the noise level when parameter estimation becomes corrupted.
- Use the chosen data representation signal setup for the addition of different levels of noise up to the threshold of the generator signal in the simulation to estimate the impact on the addition of noisy data to the data sets for NN model sustainability.

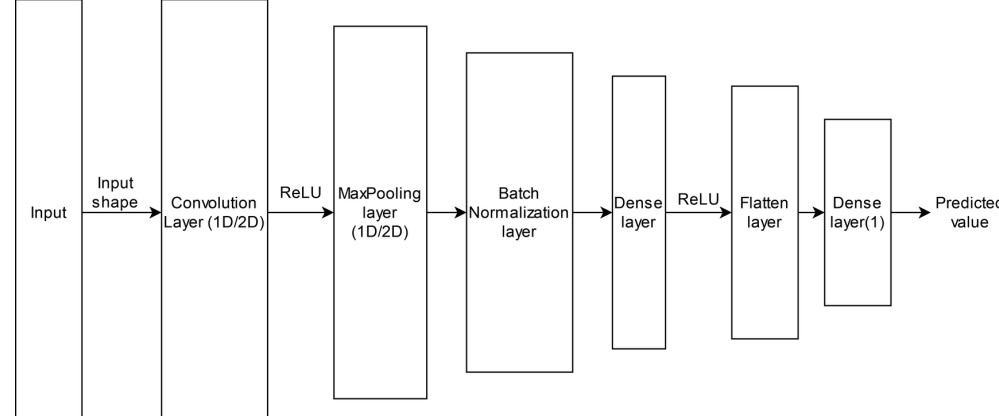

**Figure 1.** Initial architecture of the model trained using synthetic data.

### 2.3. Neural Networks

*Linear regression problem*

Additional data were synthesized to solve problem (E), namely, to estimate parameter values with the noise added both to the synthesized output and generator signal during the simulation, since all of the previous data used were ideally synthesized. Data sets were created with two varying parameters ($\epsilon_r$ and $\tan\delta$), and the noise was added to the clear signal data set as different Gaussian noise levels:

- Noise level of 10% to whole signal;
- Noise level of 20% to whole signal;
- Noise level of 20% to parts of signal (noise spikes).

Then, using the established noise level threshold data sets of noise added to the radar signal, DCT decomposition coefficients were created in order to evaluate the impact of the noise level on predicted parameter values. A separate data set with no noise added to the generator signal was created for comparison of the results. The data sets included data with no noise added, and with noise levels of 5%, 10%, 15%, and 20% added to the signals.

## 3. Results and Discussion

Due to the extensive nature of the study, some of our results are described in detail in separate articles:

- Conformance of the developed mathematical model of EM propagation through a layered medium to the simulation could be improved (please see Conclusion in [39]);
- Many NN architectures were not tested [29];

### 3.1. Development of the Multistatic UWB IR

During the study, there was an upgrade applied to the existing UWB IR radar:

- Designing the optimized antennas for frequencies above 2 GHz (Figures 2 and 3);
- Designing the antenna switch that ensures good isolation between channels (for signals up to 3 GHz in frequency, $S21 > -1.5$ dB when the channel is on, and $S21 < -43$ dB when the channel is off);

- Redesigning the firmware of the existing radar in order to support multistatic radar data flow;
- Designing the human–machine interface software of the upgraded radar.

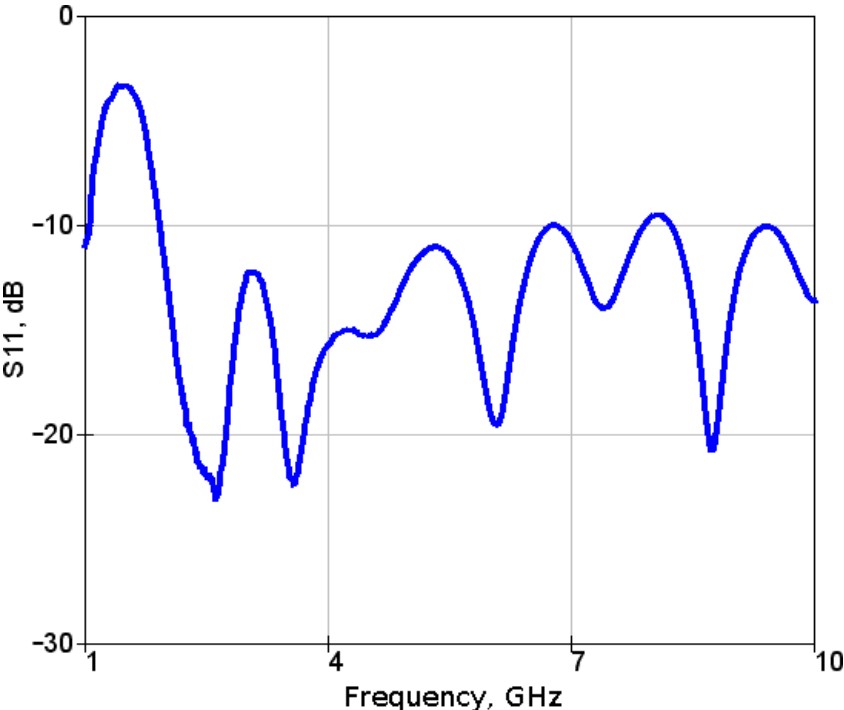

**Figure 2.** Measured S11 of the developed wideband antenna.

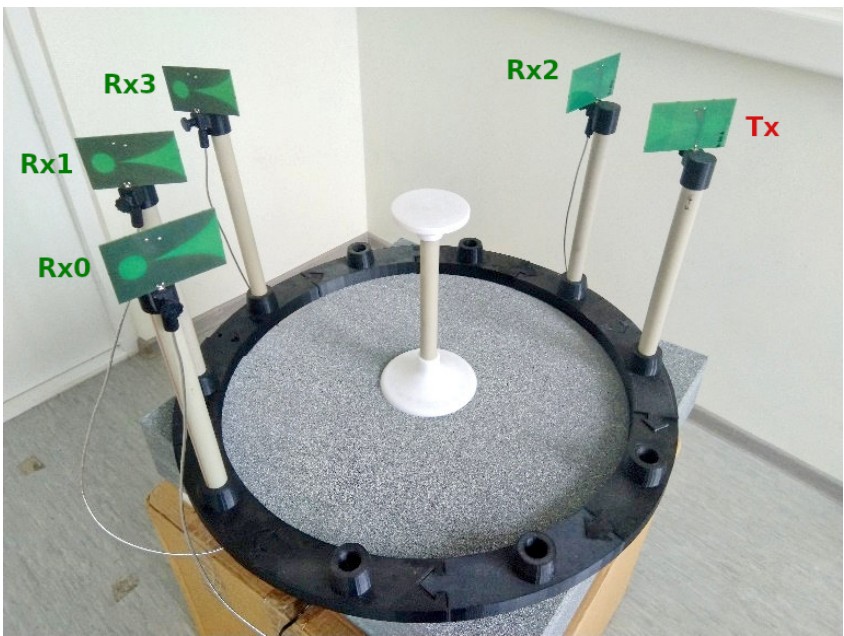

**Figure 3.** Multistatic UWB IR experimental measurements setup: Tx-transmitting antenna, from Rx0- to Rx3-receiving antennas.

The parameters of the upgraded radar are listed in Table 1.

All the previously listed hardware components were assembled to create the measurement setup (Figure 3). The radar and antenna switch boards were placed below the broadband EM wave absorber foam block in order to reduce interference from the electronics; the foam block also reduced the reflections of the whole measurement setup. The antenna holder base circle is composed from 10 modular, connectable, 3D-printed PLA

plastic sections with locations for antenna insert stands on each section. Such segmented construction allows flexibility in antenna placement so that different radar configurations can be tested easily:

- Pseudo-monostatic radar, in the case when the receiving antenna is placed next to the transmitting antenna;
- Bistatic radar, in the case when the receiving antenna is not placed next to the transmitting antenna;
- Multistatic radar, in the case when multiple receiving antennas are used.

The 3D-printed stand for objects under analysis is placed in the center of the circle. Inner diameter of the antenna holders circle is 60 cm; stand with an object is at closest about 25 cm from the antennas. Antenna fixtures along with object stand are raised about 40 cm from the base in order to reduce reflections from it.

**Table 1.** Parameters of the upgraded radar.

| Property | Value |
|---|---|
| Size, mm | $60.5 \times 98.5$ |
| Input voltage PoE or AUX, V | 12 to 48 |
| Antenna connector type | SMA |
| Pulse repetition frequency, MHz | 1.38 |
| Time window, ns | 18 to 50 |
| Sample count | 400 to 4096 |
| Equivalent sample rate, GSps | 8 to 130 |
| ADC sample size, bits | 8 to 12 |
| Communication interface | Ethernet |
| Pulse bandwidth, GHz | 2 to 4.5 |
| Pulse voltage on 50 Ohm load, V | 6 |
| Pulse rise time, ps | 55 |
| Pulse half width time, ps | 98 |
| Output impedance, Ohm | 50 |
| Dynamic range of the receiver, dB | 60 |
| Input noise level of the receiver, mV | 1 |

*3.2. The Developed Model for Determination of Dielectric Parameters of Objects Using UWB Impulse Radar*

The EM wave propagation model was developed for the case shown in Figure 4: the three-layered medium is made from theoretically tree different dielectrics. Practically, the medium before and after the object under analysis with thickness $x_2$ and relative permittivity $\varepsilon_2$ is air, resulting in $\varepsilon_1 = 1$ and $\varepsilon_3 = 1$. Other denominations in the picture are

- $E_0$—The unknown magnitude of the wave that comes from the transmitting antenna $A_T$;
- $E_1$, $E_2$—Waves reflected from the boundaries between front and back of the object measured by the receiving antenna $A_R$;
- $E_3$—The wave that has passed through object;
- $\tan \delta_2$—The loss tangent of object;
- $x_0$—The distance from the antennas to the object.

The model is described in detail in [40]. The testing of the model was conducted in order to determine whether it could be used in EM wave propagation simulation for synthetic data generation.

*3.3. Model Conformation Using Synthetic Data*

The model conformational analysis was performed using synthetic data, since data generation for NN evaluation was planned using EM field solver OpenEMS. The simulated scene was similar to the one shown in Figure 4: two identical antennas (transmitting $A_T$ and receiving $A_R$) were placed at a distance of 40 mm and pointed towards a wall away from the dielectric under analysis; such a simulation setup represents the particular practical

experiment setup established in the laboratory. The model conformation is described in depth in [39]. Conformance analysis can take a whole year to develop tools for analysis and perform necessary simulations and calculations; hence, we used a simplified analysis method, which showed weak conformance (around 40%) to the developed model. This can be explained by the nature of the model: it consists of two non-linear transcendental equations with very sensitive solutions, leading to high uncertainties in the solver, not even taking into account high error in the determination of the position in time of the reflected signals. Accordingly, the conformation tests proved that a mathematical model for material properties estimation is hard to develop, and that the material properties estimated according the developed model can be unreliable [39].

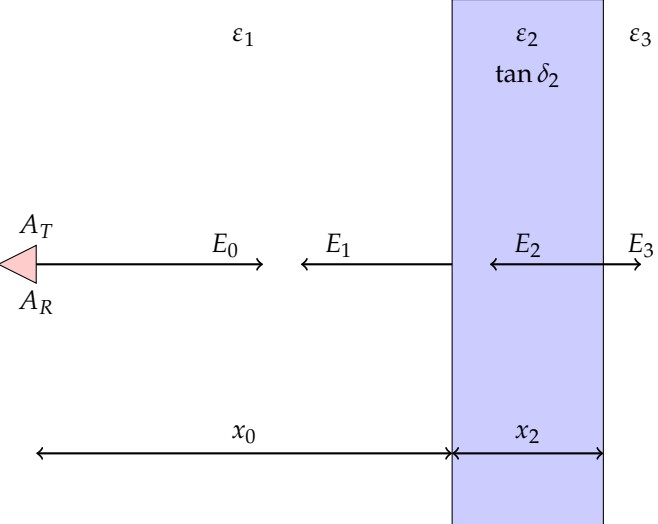

**Figure 4.** Three-layer model with an object representing one layer of dielectric medium.

### 3.4. The Synthetic Data Generation

The reasons for synthetic data generation are

- The comparison of the simulation output with the experimental measurement results;
- The validation of the developed EM wave propagation model;
- The generation of data for the training of the NN and development of NN architectures.

The data generation at different stages of project was performed using different signals and different scene setups, as described below.

*Simulated Scenes*

Two types of scenes were simulated:

- The dielectric block wall scene was simulated imitating monostatic and bistatic radar configurations in order to

  - Validate the developed EM wave propagation model;
  - Compare synthetic data with experimental measurements;
  - Generate data sets for NN training and evaluation.

  Simulation of this scene took a lot of time since the dimensions of the scene were around 70 wavelengths, and real radar antennas in the simulation were used, which lead to a very large mesh size. The front of the wall was up to $4.4 \times 4.4$ m (in order to shift the reflection from the wall vertexes outside the time window when the signal is reflected from the bricks of the back wall); the physical parameters of the scene were variable:

  - Distance to the wall, 350 to 700 mm;
  - Thickness of the wall, 200 to 500 mm;
  - Relative dielectric permittivity of the wall, 1 to 6;
  - Dielectric losses of the wall, 0 to 0.1.

- The dielectric ball (13 cm in diameter) was simulated imitating real bistatic and multistatic radar scene configurations in order to rapidly generate NN training data. This scene was simulated instancing eight simple half-wave dipole antennas in close proximity to the ball in order to minimize the size of the scene's mesh and obtain training data as quick as possible. Antennas were equidistantly placed on the square, which was 200 × 200 mm in size. The physical parameters of the scene were variable:
  - Relative dielectric permittivity of the ball, 1 to 6;
  - Dielectric loss of the ball, 0 to 0.25.

*Signals Used in Simulations*

Different signals were tested to excite the scene antennas in OpenEMS. In the end, the following signals were used:

- The Gaussian signal (Figure 5a), since it is continuous and its spectrum due to the Fourier transform properties is also Gaussian, and therefore also continuous. The spectrum of the used Gaussian signal is similar to the spectrum of the radar signal used at the beginning of the project, so this signal was used during simulations of the first data sets for NN training, since the model of the developed EDI radar was not yet developed;
- The sine signal truncated to a single period (Figure 5b), since it provided better results in terms of finding the proper positions in time when analyzing the reflected signals in the developed model of EM wave propagation;
- The signal of the EDI radar (Figure 5c), since it should provide simulation results that were the most similar to the experimental results. The signal of the EDI radar was modeled using the 66 most powerful components of the discrete cosine transform from the recorded EDI radar signal. This signal was used for the generation of synthetic data for NN training and evaluation. This signal was also used for noisy transmitted signal generation, varying, by up to 20%, the DCT coefficient values in order to simulate the imperfectly matching radar signals.

Additional noise on demand was added to the received signals in order to represent noisy signals received by the radar.

Spectra of the signals are shown in Figure 6: as can be seen, the signal of the EDI radar is close to perfect for measurements, since the variation in power is less than ±1.3 dB in the 0.5 to 5 GHz band.

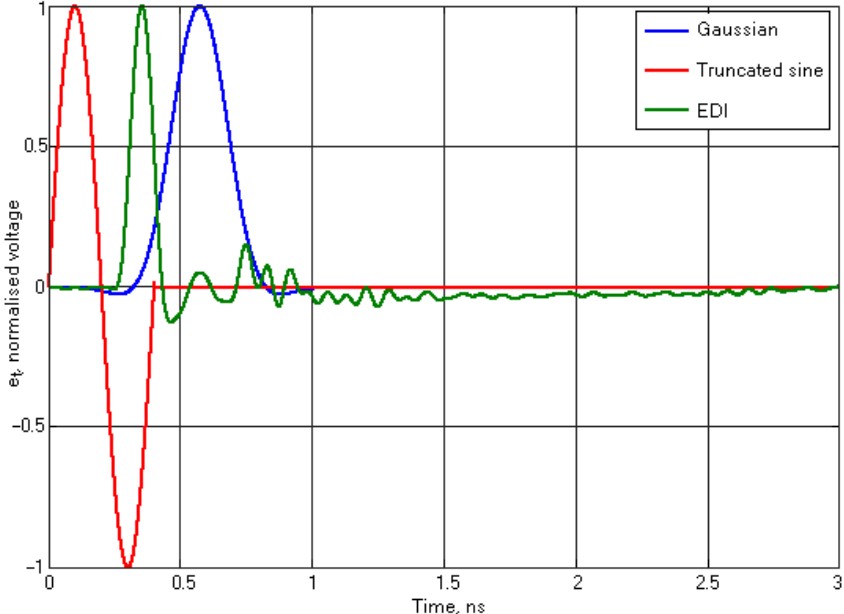

**Figure 5.** Normalized signals used in simulations: (a) blue line—Gaussian signal *SetGaussExcite*(*FDTD*, *1e9*, *2.5e9*); (b) red line—truncated sine signal; (c) green line—Signal of EDI radar.

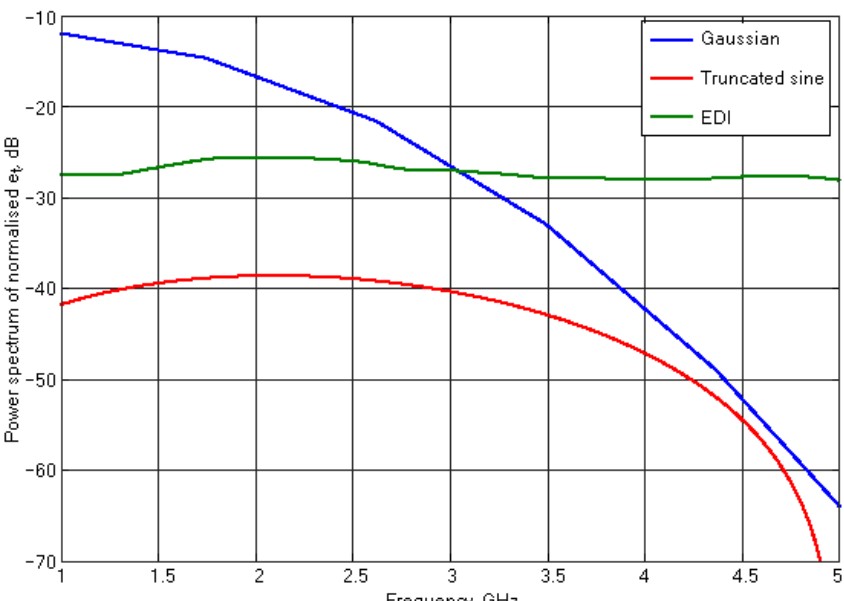

**Figure 6.** Power spectra of normalized simulation signals: (a) blue line—Gaussian signal *SetGaussExcite*(*FDTD, 1e9, 2.5e9*); (b) red line—truncated sine signal; (c) green line—signal of EDI radar.

### 3.5. The Synthetic and Experimental Data Comparison

In order to evaluate the correctness of the simulated signals, the real life experiment and corresponding simulations were generated, in which two concrete bricks with dimensions $620 \times 620 \times 77$ mm and $620 \times 620 \times 95$ mm at distances 500 and 700 mm were radiated in a pseudo-monostatic radar configuration, and reflected signals were registered for comparison; then, the registered signals were correlated.

The correlation ranged from 38% to 79%; the sample results are shown in Figure 7. Comparison shows that simulated signals contain more high-frequency components than the real measured signals. This can be explained by the fact that

1.  In order to make calculations faster, the simulation was generated using simplified antenna model possessing

    - A frequency-independent antenna and substrate dielectric losses equal to 0.026;
    - An infinitely thin perfect electric conductor of antenna surfaces;

2.  The radar signal was measured on a 50 $\Omega$ load instead of actual antenna impedance (Figure 8), due the lack of the active high-impedance RF probe for frequencies 2 to 10 GHz.

The comparison shows that the simulation is as accurate as the simulation model: e.g., the high-frequency components in the simulation reduced in amplitude if the concrete was simulated to experience losses (up to $\tan \delta = 0.2$ in Figure 7). In order to improve the conformance of the simulated signals to the real life signals, the simulation has to be seriously redesigned to eliminate the counted simplifications; this was not conducted in the present study due to project implementation time constraints.

### 3.6. Parameter Estimation Using the NN

With the regression problem, the main task for the CNN model's training outcome is to predict the value of the proposed parameter with as little error as possible. The prediction is based on the training of the model on synthesized data signals of various representations. Although the model might converge without feature normalization, it makes training more stable. Furthermore, the background signal was removed from all the signal data (signal with $\epsilon_r = 1$ was recorded as air's relative permittivity).

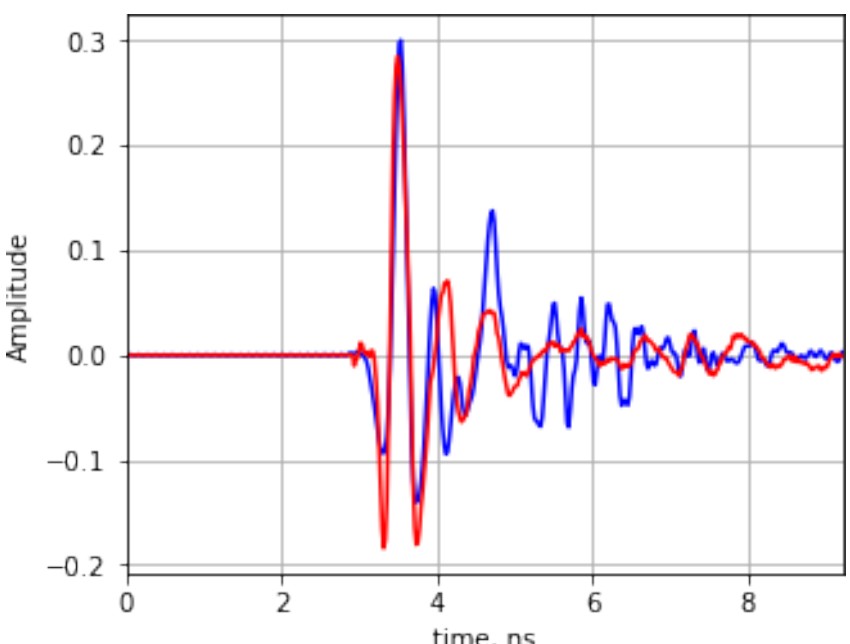

**Figure 7.** The simulated (blue) and measured (red) radar signals (shifted due to measurement offset) from a cement brick of thickness 95 mm at a distance 500 mm (correlation 79%).

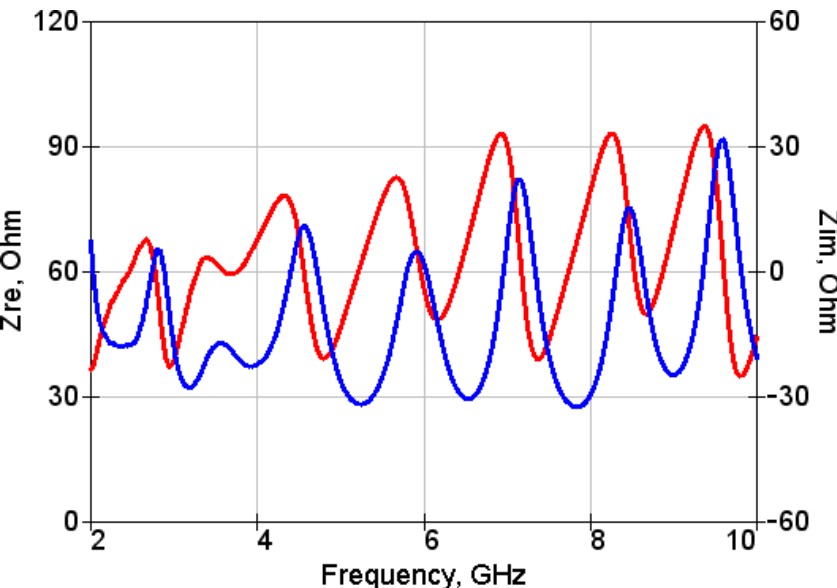

**Figure 8.** The real (red, left axis) and simulated signals (blue, right axis) of developed antenna (Figures 2 and 3) impedance.

At first, the regression problem solution was introduced to the neural network with the data sets comprising the single varying parameter to research the impact of different loss functions on the prediction accuracy. The MSE loss function showed the best performance among the other functions; therefore, it was used in further research. The loss function determination was achieved with single-parameter-varying data sets. The improvement in accuracy was established by prediction error decrease with preprocessing actions. Then, the two-parameter-varying data sets were introduced to increase the complexity and to perform further object and material structure estimation, as the $\epsilon_r$ prediction error was as low as $(0.12 \pm 0.01) \cdot 10^{-3}$ with single-parameter-varying data sets.

*Setup and data representation impact on the evaluation of parameters*

The results achieved for the value prediction from the two-parameter-varying data sets are presented in Tables 2 and 3, with the best performance being achieved by the radar multistatic configuration.

**Table 2.** Relative permittivity $\epsilon_r$ parameter estimation with different representations.

| Repres. | Signal | Error in Prediction $\times 10^{-3}$ | Training Time (s) | Parameters |
|---------|--------|--------------------------------------|-------------------|------------|
| Time domain | Refl. | $170.37 \pm 4.81$ | 9.12 | 13625 |
| | Trans. | $195.12 \pm 3.99$ | 8.96 | 13625 |
| | Bist. | $123.39 \pm 2.67$ | 19.79 | 25913 |
| | Multist. | $60.61 \pm 1.96$ | 24.83 | 38201 |
| PCA | Refl. | $2.82 \pm 0.08$ | 2.8 | 1619 |
| | Trans. | $3.25 \pm 0.09$ | 2.98 | 1619 |
| | Bist. | $2.59 \pm 0.05$ | 2.86 | 1619 |
| | Multist. | $1.77 \pm 0.03$ | 3.39 | 1619 |
| Amplitude spectrum | Refl. | $107.77 \pm 3.41$ | 4.72 | 5385 |
| | Trans. | $29.65 \pm 1.22$ | 5.82 | 5385 |
| | Bist. | $28.19 \pm 0.36$ | 8.91 | 9456 |
| | Multist. | $24.23 \pm 0.23$ | 11.27 | 13569 |
| Phase spectrum | Refl. | $50.67 \pm 2.66$ | 3.86 | 5153 |
| | Trans. | $19.08 \pm 0.38$ | 2.99 | 5153 |
| | Bist. | $16.37 \pm 0.29$ | 4.68 | 9153 |
| | Multist. | $17.28 \pm 0.31$ | 5.99 | 13569 |
| STFT | Refl. | $12.38 \pm 0.21$ | 21.93 | 39569 |
| | Trans. | $6.43 \pm 0.06$ | 26.71 | 39,569 |
| | Bist. | $6.06 \pm 0.05$ | 48.57 | 79,249 |
| | Multist. | $3.78 \pm 0.08$ | 72.02 | 89,285 |

The CNN had more information about the signal changes from different perspectives, as the setup used reflected, transmitted, and side signals. Nevertheless, in some cases, one of the signals decreased the performance of the multistatic setup. With the usage of the early stopping call-back function, the model stops training when the learning curve enters the plateau and validation loss is not decreasing. The best training weights were chosen to generate predictions based on the evaluation data set.

From the data representation perspective, the worst performance was with the time domain data (filtered raw data). With the data preprocessing, higher prediction precision was achieved:

- For frequency domain representations, the phase spectra data sets for both estimated parameters showed better results as the data were more informative than in the amplitude spectra data sets. The prediction error is 1.41 and 1.76 times smaller with phase spectrum than the amplitude spectrum for $\epsilon_r$ and $\tan\delta$, respectively (multistatic setup). The spectrum component number impact was also studied, where the latter components served as noise and decreased the performance of the NN training;

- The STFT preprocessing showed improved results compared to the previous data representations due to the ability of the method to extract the most valuable information from time and frequency domain data. With STFT, parameter prediction error is 4.57 units smaller for $\epsilon_r$ and 1.07 times smaller for $\tan\delta$ compared to the phase spectrum results for multistatic setup. As the data sets of STFT are presented as 2D images with additional features of RGB color intensity, the number of input data dimensions increased. Therefore, the number of trainable parameters increased by 6.79 to 61.97 times depending on the antenna setup, which led to a training time increase;

- Finally, the PCA preprocessing showed best prediction error results for both estimated parameters, with $(1.77 \pm 0.03) \cdot 10^{-3}$ for $\epsilon_r$ and $(141.03 \pm 3.31) \cdot 10^{-6}$ for $tan\delta$. The performed tests indicate that simplification of the data by feature vector reduction was achieved. Therefore, the training time is reduced (by 21.24 to 38.19 times compared to the STFT on multistatic setup data) along with trainable weight parameters (by 55.15 to 57.0 times). The PCA method also allowed us to train the neural network model with the same number of training parameters for every setup, as the number of principal components remained the same. The method itself was implemented via the *scikit-learn* machine learning library to perform such dimensional reduction. The PCA class was used to fit and transform the raw input data.

**Table 3.** Dielectric loss *tanδ* parameter estimation with different representations.

| Repres. | Signal | Error in Prediction $\times$ $\mathbf{10^{-3}}$ | Training Time (s) | Parameters |
|---|---|---|---|---|
| Time domain | Refl. | $1711.83 \pm 34.88$ | 17.28 | 16227 |
| | Trans. | $1231.36 \pm 20.11$ | 12.95 | 16227 |
| | Bist. | $410.81 \pm 11.32$ | 11.2 | 7337 |
| | Multist. | $240.05 \pm 5.07$ | 15.59 | 10921 |
| PCA | Refl. | $170.91 \pm 5.31$ | 2.7 | 1619 |
| | Trans. | $181.88 \pm 4.11$ | 2.38 | 1619 |
| | Bist. | $192.76 \pm 3.77$ | 2.69 | 1619 |
| | Multist. | $141.03 \pm 3.31$ | 2.9 | 1619 |
| Amplitude spectrum | Refl. | $200.01 \pm 5.1$ | 19.62 | 3313 |
| | Trans. | $3200.1 \pm 505.09$ | 23.17 | 3313 |
| | Bist. | $1310.57 \pm 207.31$ | 29.98 | 6033 |
| | Multist. | $990.22 \pm 139.35$ | 54.35 | 8769 |
| Phase spectrum | Refl. | $990.1 \pm 110.37$ | 12.06 | 689 |
| | Trans. | $740.9 \pm 60.21$ | 8.34 | 689 |
| | Bist. | $751.33 \pm 61.01$ | 12.79 | 1089 |
| | Multist. | $562.41 \pm 40.1$ | 17.67 | 1489 |
| STFT | Refl. | $532.23 \pm 36.5$ | 21.93 | 30839 |
| | Trans. | $362.65 \pm 20.34$ | 26.46 | 30839 |
| | Bist. | $437.71 \pm 40.1$ | 77.46 | 61559 |
| | Multist. | $521.32 \pm 49.21$ | 110.77 | 92279 |

*Noise impact on the parameter estimation*

With the achieved results, the background noise removal, data normalization, and PCA preprocessing enhanced results were achieved for the estimation of the object material and structure. Only transmitted channel signals were used (in the clearly monostatic radar case) to decrease the calculation time. Based on the knowledge gained during the study, transmitted channel (monostatic radar) signals enhanced the parameter value prediction compared to the reflected signal in bistatic radar configuration. Data sets were tested with noise added both to the initial clear synthesized signals and excitation signals of the simulator.

Data sets with noise added to the initial clear signals were used to test the NN model and study the impact on the prediction accuracy, with various noise levels added after the data synthesis. The higher the noise level, the harder it is to predict the accuracy of the parameter's value. Training time increases together with the necessary epoch number to reach the prediction error. Noise addition only to parts of the signal has less impact on the prediction accuracy.

Even with the SNR of 5only at the random signal parts, high precision was obtained with only $(69.39 \pm 8.9) \cdot 10^{-3}$ and $(1.14 \pm 0.01) \cdot 10^{-3}$ average prediction error of the relative permittivity and dielectric loss accordingly. Therefore, the noise addition to parts of the

signal will not be further studied. The obtained results are shown in the Table 4. For data sets with noise added to the generator signals, two scenes were evaluated to estimate the ability to predict the parameter value. The first scene consisted of data sets with different noise levels, both in training and evaluation data sets. The second scene consisted of UWB signal data with no noise added to the generated signals, while the NN model evaluation was performed on data with different noise levels.

**Table 4.** Estimation of $\epsilon_r$, $tan\delta$ with different noise levels added.

| Represent. | SNR | Error in Prediction $\times\ 10^{-3}$ | Training Time (s) | Average Epochs |
|---|---|---|---|---|
| Time domain ($\epsilon_r$) | None | $19.72 \pm 0.62$ | 5.27 | 6.93 |
| | 0.1 | $89.84 \pm 16.46$ | 5.32 | 14.41 |
| | 0.2 | $722.29 \pm 141.22$ | 6.11 | 18.47 |
| | 0.2 (spikes) | $69.39 \pm 8.9$ | 6.55 | 20.73 |
| Time domain ($tan\delta$) | None | $0.39 \pm 0.04^{-3}$ | 6.1 | 19.8 |
| | 0.1 | $1.19 \pm 0.04^{-3}$ | 6.48 | 20.54 |
| | 0.2 | $5.24 \pm 0.24^{-3}$ | 6.31 | 23.47 |
| | 0.2 (spikes) | $1.14 \pm 0.01^{-3}$ | 6.65 | 21.13 |

Therefore, such evaluation shows

1. Whether the parameter evaluation is achievable with noisy data;
2. Whether parameter prediction is possible on noisy data while training with synthesized data without any noise added;
3. How the noise level affects the accuracy of the parameter prediction.

The noise added to the used data sets in Table 5 was created in OpenEMS by adding random noise to the EDI radar signal used for excitation of the antennas. Testing such types of noise was conducted due to the hardware often not being ideal. Therefore, thermal or flicker noises might occur along with other noises such as crosstalk, interference, capacitive coupling, or ground loops.

The results in Table 5 show that it is important to include in training data with noise: the case when NN was trained only on the data set without noise shows worse parameter estimation performance than the case when NN was trained on data with added noise. With the increase in the noise level, prediction error increases due to noise influence on the signal data stability. The prediction errors are satisfactory for the relative permittivity estimation of 0.05, 0.1, and 0.15 noise levels. On the other hand, the loss tangent predictions are chaotic and exhibit large prediction error deviations. Therefore, data with noise is preferable to the training data set as additional information.

The results on the synthesized UWB signal data with various noise levels are shown in Figure 9 for dielectric loss parameter estimation (identical behavior was established with relative permittivity). The relative permittivity showed better prediction results with higher accuracy, while loss tangent values were harder to predict (as with previous accuracy performance evaluations). Nevertheless, accuracies up to $(0.71 \pm 0.01) \cdot 10^{-3}$ for relative permittivity and $(90.0 \pm 0.05) \cdot 10^{-6}$ for the loss tangent were achieved with the CNN model, MSE loss function, and PCA preprocessing method. This enables the possibility of importing more varying parameters for further estimation of object material and structure estimation. Data with noise addition must be included in the data sets of synthesized UWB signals due to the fact that real life data is often influenced by the surrounding noise sources as well as hardware imperfections.

**Table 5.** Parameter estimation error with noise added to the generator signal.

| Represent. | SNR | Error in Prediction $\times 10^{-3}$ | Data Used |
|---|---|---|---|
| PCA $(\epsilon_r)$ | 20 | $0.71 \pm 0.01$ | Both training and evaluation on data with noise |
| | 10 | $1.03 \pm 0.01$ | |
| | 6.67 | $2.84 \pm 0.02$ | |
| | 5 | $10.19 \pm 0.09$ | |
| PCA $(tan\delta)$ | 20 | $0.09 \pm 0.5^{-3}$ | |
| | 10 | $0.13 \pm 0.9^{-3}$ | |
| | 6.67 | $0.28 \pm 1.2^{-3}$ | |
| | 5 | $0.74 \pm 2.4^{-3}$ | |
| PCA $(\epsilon_r)$ | 20 | $122.73 \pm 21.14$ | Training on data without added noise, evaluating only on data with noise |
| | 10 | $148.93 \pm 25.81$ | |
| | 6.67 | $229.98 \pm 55.51$ | |
| | 5 | $772.45 \pm 195.42$ | |
| PCA $(tan\delta)$ | 20 | $5.82 \pm 0.09$ | |
| | 10 | $7.55 \pm 0.13$ | |
| | 6.67 | $10.43 \pm 0.27$ | |
| | 5 | $18.87 \pm 0.41$ | |

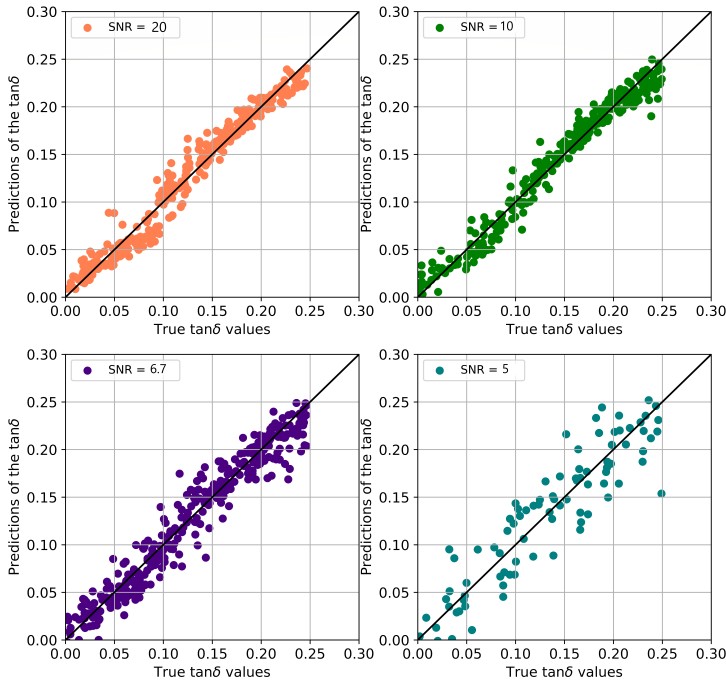

**Figure 9.** Dielectric loss estimation of data with different noise levels.

### 3.7. Monostatic and Multistatic Radar Application for Material and Structure Recognition

One of the project hypotheses declares that the multistatic radar has to have advantages over the monostatic radar in object material and structure recognition with AI. This topic was studied, and is described in detail in [29,30]. Outcomes summary:

- The object detection accuracy was worst in the case of the monostatic radar [29];
- In the case of the multistatic radar, the detection accuracy increased by 6% in comparison to monostatic or bistatic radar cases;
- Switching from monostatic to multistatic radar increases the detection accuracy for a variety of tested NNs and for simple NN architectures, and reduces the mistakenly classified object count up to four times [30].

These findings can by explained by additional information carried in signals that have been passed through or scattered from objects [29].

*3.8. Comparison of Different Neural Network Architectures in UWB IR Signal Processing*

One of the most important problems, as defined previously, is the choice of proper NN architecture for UWB IR signal processing. This topic was studied and is described in detail in [30]. The essential outcomes are as follows:

- The difference in proper waste detection accuracy for simple NN architectures (multi-layer perceptron and convolution NN) and dense NN architectures (Transformer and ResNet) in the case of the monostatic radar is up to 1.9% in favor of dense NN architectures; at the same time, this means a difference in counts of mistakenly recognized objects of up to four times;
- In the case of simple NN architectures switching from monostatic to multistatic radar increases the detection accuracy up to 1.8%;
- In the case of most dense NN architectures, switching from monostatic to multistatic radar increases the detection accuracy only by a fraction of a percent.

## 4. Future Research

There are multiple directions where the findings from the present study can be developed:

- The conformance of the developed mathematical model of EM propagation through a layered medium to the simulation can be improved (please see Conclusion in [39]);
- Many NN architectures were not tested [29];
- Wavelets for data preprocessing were not tested;
- The classification of real waste could be improved using radar data and visual camera data fusion in NN;
- The experimental setup can be developed into industrial waste sorting equipment: the antennas could be placed around the conveyor belt pointing towards a single piece of waste on a belt at a 30 to 45 degree angle; in order to avoid EM interference with other equipment in waste sorting centers, the area surrounding the antennas will probably need to be shielded; the radar itself can be made airtight and powered using Power over Ethernet.

## 5. Conclusions

In our study, the existing monostatic radar was upgraded to the multistatic radar with the help of software, firmware, and switches of the antenna signal: different transmission and reception antenna configurations are possible (16:1, 4:1, 1:1, 4:4, 1:4, 1:16).

A new optimized antenna was designed for the frequency band 2..10 GHz with the help of free FDTD simulation software OpenEMS [26].

In OpenEMS, simulations generated synthetic data that can be used to study and develop optimized artificial neuron network structures for the estimation of dielectric material properties.

The application of the developed mathematical model of EM wave propagation through dielectrics can lead to incorrect estimation of dielectric material properties due to errors in determination of reflected signal position in time and the sensitive equations of the model. Instead, monostatic UWB IR in combination with NN was successfully used for the determination of material properties, such as thickness, dielectric permittivity, and losses, with improved, but limited, accuracy; multistatic radar configuration of such a system increases the estimation accuracy of given properties. As it is often time-consuming and requires expensive equipment, synthetic data gathering was used to reduce the time required for sample collection. Therefore, synthetic data were gathered in small steps to cover the mentioned range of values, with steps of up to $1.0^{-3}$. A non-destructive method was used that allowed us to use the data sets for the CNN to solve the regression problem and estimate one or two varying parameter values, and thus to evaluate the EM properties of the material and internal structure of the objects.

With the methodology proposed, the MSE loss function showed a lower prediction error than other loss functions on the single-parameter-varying data set. The two-parameter-varying data sets were introduced with different data representations and preprocessing. Distinct data setups were used to confirm the statement in the paper related to the study project [29]. In almost all cases, the multistatic setup showed the best performance.

In the case of the representations, the PCA method and STFT showed favourable performance in parameter value estimation (Tables 2 and 3). While STFT is more robust, PCA enables the reduction in input vector (feature) size that simplifies the NN architecture. This is indicated in Tables 2 and 3. The scale of trainable parameter reduction ranges from 19 to 57 times. The training time reduction ranges from 9.5 to 38 times. The range depends on the setup used during the training process. PCA preprocessing allowed us to reduce training time, improve prediction accuracy, and enhance the possibility of implementing regression problem-solving techniques on a device (microcontroller) or, in other words, on the edge.

With noise-added data, the results show that at the noise spikes, data introduction did not affect the prediction capability of the NN model (see Table 4). In comparison, the Gaussian noise of 0.2 level predicted parameters chaotically with differences of 0.63 for the $\epsilon_r$ and $4.1 \cdot 1.0^{-3}$ for the $tan\delta$ prediction. The 0.2 noise level, therefore, was chosen as the top margin for noise introduction to the generator signal in the simulation environment.

With the accuracy achieved by implementing the proposed methodology, data preprocessing, and representations, object material and structure estimations are possible. The NN opens possibilities to extract information about the material structures for further evaluation and comparison of the parameters. It opens possibilities to implement more parameters for evaluation and research on both synthesized and real-life data.

It has been proven that in order to speed up the optimal NN model development for real-life applications, it is feasible and useful to develop NNs using synthetic data and to tune NN architecture using real data. With the proposed methodology, rapid NN model creation is possible, with options to reduce input features and enhance the model with data preprocessing. This may lead to training time reduction, performance improvements, and complexity reduction in the NN. The solution is usable in various fields, such as concrete structure evaluation, or parameter, shape, and size estimation with different material classifications.

Our study shows that NNs can be used to determine the properties of the materials from radar measurements even if direct and reflected signals are not resolved in time (reactive near field).

**Author Contributions:** Conceptualization, M.G.; methodology, M.G. and V.A.; software, G.G., R.M., H.E. and K.M.G.; validation, G.G., R.M. and K.M.G.; formal analysis, V.A., G.G., R.M. and K.M.G.; investigation, V.A., G.G., R.M. and K.M.G.; resources, J.S., E.L., H.E. and D.L.; data curation, R.S., R.M. and K.M.G.; writing—original draft preparation, G.G. and R.M.; writing—review and editing, G.G. and R.M.; visualization, G.G. and R.M.; supervision, M.G.; project administration, R.S.; funding acquisition, M.G. All authors have read and agreed to the published version of the manuscript.

**Funding:** This study was funded by the Latvian Council of Science, project No. lzp-2020/2-0270.

**Institutional Review Board Statement:** Not applicable.

**Informed Consent Statement:** Not applicable.

**Data Availability Statement:** Used data sets for NN training can be obtained contacting info@edi.lv and referencing project MIRSA.

**Conflicts of Interest:** The authors declare no conflict of interest. The funders had no role in the design of the study; in the collection, analyses, or interpretation of data; in the writing of the manuscript; or in the decision to publish the results.

## Abbreviations

The following abbreviations are used in this manuscript:

EDI     Elektronikas un datorzinātņu institūts (Institute of Electronics and Computer Science)
UWB     Ultra-wideband
IR      Impulse radar
EM      Electromagnetic
NN      Artificial neuron network
CNN     Convolution neural network
FDTD    Finite difference time domain
PCA     Principal component analysis
STFT    Short-time Fourier transform
MSE     Mean squared error

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
