# Peer review of "Evaluation of Materials and Structures with a Multistatic Ultra-Wideband Impulse Radar: A Concept Validation"

_applsci, doi:10.3390/app13031636_

Round 1

Reviewer 1 Report

The project is very interesting. The authors provide a concept implementation of materials and structures classification using multi-static ultra wide band impulse radar. I have the following comments:

1. Please explain the merits of using multi-static radar over mono-static radar.

2. Please explain how to classify the plastic in real industrial environment using your method in the future, such as how to place the device. 

3. please explain the relationship between the signal and the relative permittivity and loss tangent precisely such that the reader need not refer other references.

4 please provide the NN structure and how to implement the PCA preprocessing.

5 There are two regression outputs. What are the weights of the two loss functionS?

Reviewer 2 Report

This manuscript describes a plastic waste sorting system that uses the multi-static ultra-wideband impulse radar with a deep learning back-end. The results seem efficacy. However, the manuscript lacks an analysis of the proposed system and it looks like an experiment report instead of an article.

There are some detailed comments.

1.      The formulas’ vectors should be written in vector format.

2.      In Fig.2, the authors should describe the UWB IR detailed.

3.      The figures, such as Fig. 4, have no description of the y label.

4.      In line 295, ‘The correlation was in range 38..79%,’, what is the meaning of ‘38..79’?

This manuscript should give more theoretical analysis before it can be accepted.

Reviewer 3 Report

(1) Use vector network analyzer test for comparison

(2) Correction of spelling errors, for example, mMultistatic radar, etc.

Reviewer 4 Report

This paper presents an evaluation report of the dielectric parameters estimation of materials with multi-static ultra-wideband impulse radar. The performance of parameter estimation using CNN training with several different representations are compared. There are several concerns that need to be addressed in order to improve the quality of the paper.

1. The abstract is not well written. It looks like a report summary of a project. It needs to be improved.

2. The state-of-the-art of the underlying research work is not sufficiently summarized. The approaches to measure and estimate the dielectric parameters of materials using UWB radar are rarely mentioned in the introduction.

3. The measurement model of the multi-static UWB radar is not clearly described. How the dielectric parameters of materials are predicted from the echo signals is also unclear.

4. The role of the CNN training played in the prediction of the parameters is not very clear. How is the training data structured? What are the inputs and outputs of the neural network? How is the network trained? What is the relationship of the CNN training with different representation methods?

5. What is the meaning of the structures mentioned in the title of the paper? Whether it is possible to obtain the structures of the object of interest is not evaluated in the paper.

Round 2

Reviewer 1 Report

This paper has been revised according the reviewer's comments. The explainations are pertinent.   

Author Response

Thank you for your comments and valuable time!

Kind regards ~
EDI project team

Reviewer 2 Report

The analysis and the figures in this manuscript are improved, and it is more convincing. Therefore, this manuscript can be accepted.

Author Response

(The authors gave the same response as above.)

Reviewer 4 Report

The manuscript has been revised. However, the description of the method can be further improved. For example, a figure illustrates the architecture of the model used to train on synthetic data is provided. However, what is the input of the model is not clear. In section 2.3, what is the features of inputs, how is it extracted? The technical content should be clearly described.
